# Building the Value Proposition of a Digital Innovation Hub Network to Support Ecosystem Sustainability

**Claudio Sassanelli** [1,*] and **Sergio Terzi** [2,*]

1    Department of Mechanics, Mathematics and Management, Politecnico di Bari, Via Orabona 4, 70125 Bari, Italy
2    Department of Management, Economics and Industrial Engineering, Politecnico di Milano, Piazza Leonardo da Vinci 32, 20133 Milan, Italy
*    Correspondence: claudio.sassanelli@poliba.it (C.S.); sergio.terzi@polimi.it (S.T.)

**Abstract:** Digital Innovation Hubs (DIHs) play a key role in bolstering European companies to overwhelm innovation barriers and drive Europe as the world's primary leader in the Industry 4.0 digital revolution; they are one-stop-shop ecosystems able to provide four main functionalities (test before investing, support to find investments, innovation ecosystems, and networking, skills and training). Even if a surge in their diffusion has been registered, their sustainability is still far from being well defined in a structured way. Several approaches and methods are available from literature to ground the sustainability plan of companies' business. Among them, the first activity to be addressed is the value proposition (VP) analysis, and the most diffused approach is the Value Proposition Canvas (VPC); this paper proposes the application of the VPC (jointly used with other methods from the VP literature) to build the VP of the HUBCAP network (supporting European small and medium-sized enterprises in the adoption of model-based design methods and tools to support cyber-physical system technologies) per each of its four main customer segments (DIHs, academic partners and research and technology organizations, technology/tool providers and technology/tool users). Results highlight the need to characterize the analysis per each of these customers, open up new opportunities to build a structured business model of the network, and constitute a basis for assessing the potential synergies with similar DIH networks. The method proposed can be applied to any other DIH or network of DIH to define their specific VP, ground the strategy to reach their sustainability, and trigger collaborations with each of the four customer segments considered in the analysis.

**Keywords:** digital innovation hub; value proposition canvas; ecosystem sustainability; digital transformation; model-based design; cyber-physical system; collaboration platform

## 1. Introduction

Digital Innovation Hubs (DIHs) play a key role in bolstering European companies to overwhelm innovation barriers and drive Europe as the world's primary leader in the Industry 4.0 digital revolution [1]; they are one-stop-shop ecosystems able to provide four main functionalities (test before invest, support to find investments, innovation ecosystems, and networking, skills and training) [2] through a certain set of assets (competences and skills, technologies, services, etc.) [3]. To be recognized, DIHs need to be part of a regional, national or European policy initiative to digitize industry, be non-for-profit organizations, have a physical presence in the region, present an updated website and have at least three examples of how they have helped a company in the digital transformation referring to publicly available information [2,4–7]. Recently, several projects have been financed by the European Commission (EC) to push them from a regional towards a wider pan-European impact [8,9], and several results have been obtained in terms of supporting models and methods to both configure their portfolios [10] and decode and build customer journeys (CJs), bridging their provision [11]; however, being not-for-profit organizations and being, so far, always financed by EC projects, the sustainability of DIHs (and of their networks)

is still far from being well defined in a structured way [12]. On the other side, from a wider perspective, several approaches and methods are available in the literature to ground the sustainability plan and the business model of organizations and companies. Among them, the first activity to be addressed is the value proposition (VP) analysis, and the most diffused approach is the Value Proposition Canvas (VPC) [13].

Among the several projects funded by the EC, the recent HUBCAP project [14] has constituted a network composed of DIHs, technology/tool providers and users, and academic partners/research and technology organizations (RTOs); this network still represents a niche because it deals with the adoption of strong technical and specialized "test before invest" model-based design (MBD) assets (models and tools) to support small and medium-sized enterprises (SMEs) in the development and adoption of cyber-physical systems (CPSs); however, even if the HUBCAP network managed to configure its service portfolio and the related typical CJs useful for the provision of these services [11], the need to enhance its capability to attract new potential stakeholders and customers with the final aim of establishing the foundation for its sustainability plan has been unveiled. Therefore, to ease such dynamics, to be able to attract more effectively new stakeholders in the innovative HUBCAP ecosystem, and to push the impact of its digital collaboration platform [15], the HUBCAP VP should be clearly detailed.

Nevertheless, in the extant literature, there is a lack of information regarding how DIH sustainability could be reached and how VP analysis in the DIH domain could be performed. To verify its effectiveness for these ecosystems, and based on the positive experience coming from the DIH4CPS project [16], this paper proposes the application of the VPC template (jointly used with other methods from the VP literature) to build the VP of the HUBCAP network. The results highlight the need to characterize the analysis per each of the four customer profiles detected, opening up new opportunities to build a structured business model of the network and constituting a basis for assessing the potential synergies with other DIH networks. Four main slogans have been developed to represent the VP of the HUBCAP DIH network, highlighting the different perspectives to be taken when assets are offered to DIHs (a trustworthy platform-driven collaboration on CPS innovation), academic partners and RTOs (a recognized network of experts to spread MBD assets adoption), tool/technology providers (a sandbox providing a tool repository able to address end-user requests), and tool/technology users (a secure and intuitive environment capable of offering a multi-user assets catalogue and related training and knowledge to test facilities).

The paper is structured as follows. Section 2 presents the literature review to ground the research (i.e., DIHs in European digitization, MBD models and tools in the CPS domain and the VPC template) and the research method adopted. Section 3 explains the research process and Section 4 the results obtained, detailing the VP per each of the DIH network's customers. Finally, Section 5 discusses the results and concludes the paper, unveiling limitations and opening room for further developments.

## 2. Literature Review and Research Method

### 2.1. Digital Innovation Hubs and European SMEs Digitization

Society's daily lives, the way people work and do business, how they comprehend and utilise the environment and natural resources, and how people interact, communicate and educate themselves are all being dramatically altered by digital technologies. DIHs assist European companies, and in particular SMEs, in deploying and employing digital technologies to enhance business/production processes, products, or services by giving access to technical knowledge, experimentation, and the opportunity to "test before investing". A successful digital transformation requires innovation services like finance guidance, training, and talent development, which they also offer [17]. Environmental considerations are also made, particularly in relation to energy usage [18,19] and reduced carbon emissions [20,21].

The characteristics of DIHs and their interactions with stakeholders have a significant impact on how SMEs approach the digital transformation process [3]. The diversity of

these ecosystems aligns with the main objectives of the EC, which include promoting their growth, expanding the network of DIHs that already exists, and establishing an integrated, flexible, and interoperable platform for DIHs from various, primarily digitally underdeveloped industries and regions. A vast pan-European ecosystem of DIHs is the end result that the EC aims to achieve. Indeed, each DIH is unique, located in specific areas, and concentrates on a variety of sectors and digital technologies. By developing, providing, and matching services jointly with other DIHs, the forthcoming European DIH (EDIH) can spark dynamics of innovation-driven cooperation. If this goal is accomplished, DIHs will not try to simultaneously fulfil all four roles; instead, they will concentrate more on the function that is most representative of them, relying on the connections and collaborations with other ecosystems [22].

In this instance, the definition of a clear VP, able to fully represent the characteristics and the assets provided by the single DIHs, can have a key role in supporting the success of such ecosystems and triggering the creation of collaborating and integrated communities specialized in different topics, industries, technologies, and approaches.

### 2.2. CPSs Supported by MBD Methods and Tools through the HUBCAP Network and Collaboration Platform

MBD is a visual method to address the design of complex control, signal and communication systems based on mathematical models [23–25]. With MBD, it is possible to prescribe the use of models through the whole development process, representing the system structure and behaviour, providing a basis for machine-assisted analysis of system properties, and supporting design decisions for technology refinement.

HUBCAP creates a collaborative environment between DIHs and SMEs, inspired by enterprise social software [26]; this environment can be accessed through a web portal [15,26], where "Access to" and "Collaborate with" services can be found. SMEs can hence access MBD assets that can offer great support in their digital transformation process, triggering a more aware test before invest phase of the technologies they want to employ in their organizations and processes. Indeed, the chance for collaboration, through the use of tools and models provided by the platform, can support companies through the whole transformation path (i.e., piloting, testing before investing, and experimenting). HUBCAP offers two catalogues of MBD assets, one composed of tools (software packages and their dependencies that enable the development, analysis and simulation of models) and the other of models (mathematical or formal abstractions of system elements [components or subsystems]). The MBD techniques offered by the DIH network have four main types (i.e., simulation, model checking, contract-based analysis and model-based safety assessment) and can be specified in specific techniques (e.g., model checking can be split into invariant model checking, linear temporal logic model checking and deadlock checking) and models (e.g., deadlock checking techniques can be divided into CHESS, mXmv, HyComp and COMPASS [detailed in several models such as GPS, Engine, etc.]). The SMEs accessing the platform can browse both of them to perform tests and experiments using a sandbox without the need to incur high investments. The sandbox of the HUBCAP platform was inspired by the results of the INTO-CPS project [27]. In it, MBD tools were integrated into a single application [28] aggregating the different models using the Functional Mock-up Interface (FMI)-based co-simulation orchestration engine [29]; this type of MBD setup offers, together with MBD tools and models, the third class of assets: operating systems (OSs), which refer to a software environment providing libraries and dependencies needed to run the tools. The main objective of the sandbox is to enable users to add MBD assets to a cart and launch them in a cloud environment, which caters to the different technicalities required and allows several users to co-develop a model by sharing the user interface of machines hosted in a cloud environment and accessible via a web browser. HUBCAP allows SMEs to have access to MBD tools and models they can utilize to foster their digitalization process; it is intended to support trial experiments that can design and develop innovative

CPS solutions; however, a solid VP must still be defined to both ground a sustainability plan and attract new customers/stakeholders.

### 2.3. VP of Innovation Ecosystems and VPC

Innovation ecosystems, often intended as boundary organizations [30], are increasingly regarded as important vehicles to create and capture value from complex VPs [31]. Dedehayir et al. [32,33] defined four main roles of innovation ecosystems (leadership roles, direct value creation roles, value creation support roles, and entrepreneurial ecosystem roles) and examined the impact of disruption on the innovation ecosystem in its entirety (i.e., the group of organizations that collaborate in creating a holistic VP for the end user). Talmar et al. [34] developed a strategy tool (called the Ecosystem Pie Model [EPM]) to map, analyse and design (i.e., model) innovation ecosystems, focusing on the constructs and relationships that capture how actors in an ecosystem interact in creating and capturing value. Instead, the most used and valuable tool able to design the characterizing VP of an organization in relation to its main stakeholders is the VPC [13] (Figure 1); it has two sides: the customer profile, clarifying the customer understanding, and the value map, describing how it is intended to create value for that customer. The customer (segment) profile describes a specific customer segment in the business model; it breaks customers down into their jobs, pains and gains:

- Gains describe the outcomes customers want to achieve or the concrete benefits they are seeking,
- Pains describe bad outcomes, risks and obstacles related to customer jobs.
- Customer jobs describe what customers are trying to get done in their work, expressed in their own words.
- The value (proposition) map describes the features of a specific VP, breaking it down into products and services, pain relievers, and gain creators:
- Gain creators describe how products and services create customer gains,
- Pain relievers describe how products and services alleviate customer pains.
- Products and services refer to a list of all the products and services around which a VP is built.

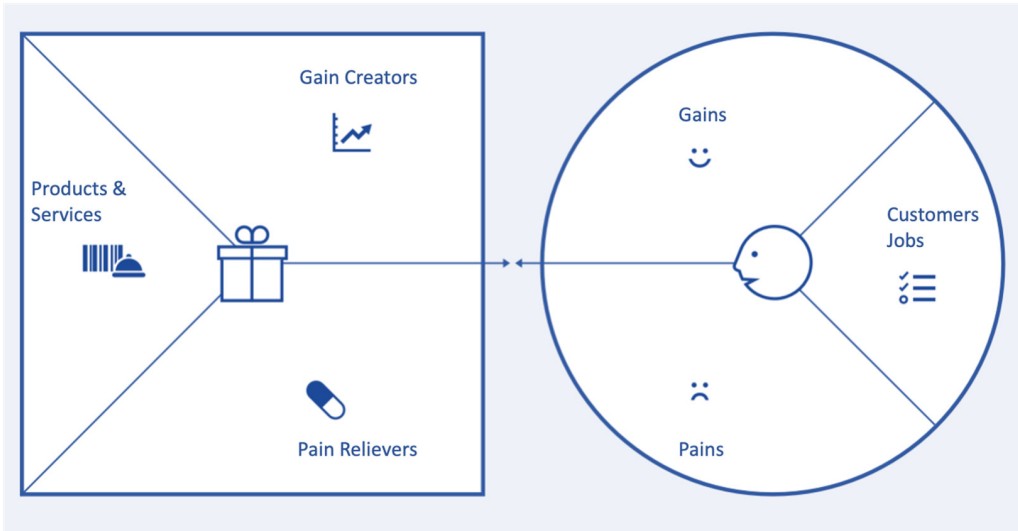

**Figure 1.** The VPC template (adapted from [14]).

The fit is achieved when the value map aligns with the customer profile (i.e., when the products and services produce pain relievers and gain creators that match one or more of the jobs, pains and gains that are important to the customer).

*2.4. Research Method*

This sub-section is aimed at describing how the VP has been built for the DIH network of HUBCAP, also based on the positive experience coming from the DIH4CPS project [16]. Four main customer segments (DIHs, academic partners and RTOs, technology/tool providers, and technology/tool users) were detected for this network. Therefore, using the Mural online collaboration platform, a workshop for each of them was conducted to explain the VPC and apply it. Each workshop lasted about 3 h, and each involved, on average, 10 people belonging to different organizations of each customer category of the HUBCAP network. In addition, more time (about 1 week) was left to them to allow each of the organizations involved to brainstorm internally to fill out the VPC template and conclude the activity. Furthermore, back-office work (about 10 h for each of the two researchers involved per VPC customer) was needed to group all the input received during the workshops. The main categories for each field of the VPC of each customer segment were defined. Then, choosing their recurrency as a driver of prioritization, groups were ranked. Afterwards, for each customer profile, the main items characterizing the VPC dimensions were defined, and fitting was done between the items referring to the dimensions composing the customer profiles (jobs, pains and gains) and those of the VP maps (products and services, pain relievers and gain creators). In particular, the resulting VP items were arranged in a pyramid-shaped template (shown in Figure 2) composed of three levels, as follows:

- Details: items contributing to a detailed description of the VP. From a customer perspective, they report the benefits coming from HUBCAP outcomes, the problems solved by them, and the arguments that prove that the organization delivering the VP is doing better than its competitors.
- Summary: two sentences synthesizing the HUBCAP offer content (answering the questions "For whom?" and "Why is it useful?").
- Title/Slogan: a sentence summarizing the VP in which the ultimate benefit of the HUBCAP offer is expressed to attract customers' attention and curiosity.

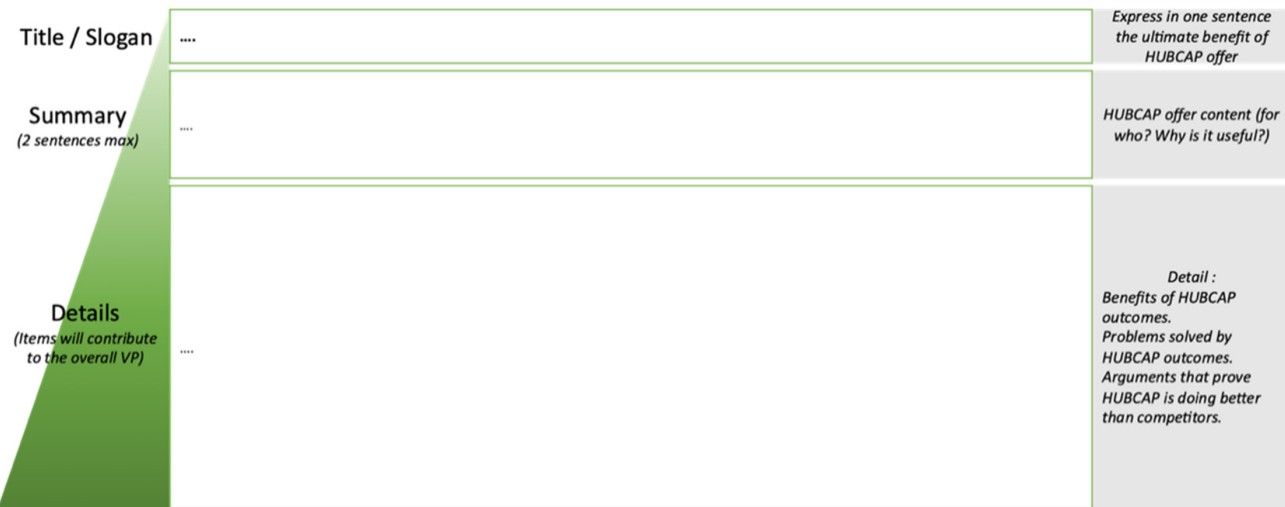

**Figure 2.** The framework used to structure the VP of each customer segment.

To translate the VP items obtained through the VPC into direct and effective statements that everyone could understand, even if they are not part of the MBD niche in the CPS domain, different theories were adopted:

- The "Value Positioning Statement" [35]: "For (target customer) who (statement of the need or opportunity), our (product/service name) is (product category) that (statement of benefit)"; it can also be simplified in the "XYZ": "We help X do Y doing Z".

- the "Jobs-to-be-done" (JTBD) theory [36]: "Action verb-Object of action-Contextual identifier".

Finally, a validation workshop was organized to check the results with the participants of the previous workshops.

## 3. Results

As mentioned in Section 1, four main customer segments were considered in the analysis of the VP of the network of DIHs providing MBD assets. Results are shown as follows. As an example, the building of the HUBCAP network VP related to the DIHs customer profile is shown step by step to better explain the research method adopted. In particular, Figure 3 illustrates the results obtained through the Mural platform, showing the detected VP items in the grey squares. After that, the two researchers gathered, grouped and ranked the input received during the workshop. The grey squares represent the VP items reported in the polished VPC in Figure 4.

Finally, Figure 5 shows the fit between the items belonging to the dimensions of the customer profile (jobs, pains and gains) with those related to the VP map (products and services, pain relievers and gain creators) per the DIH customer profile. In this figure, the items are flanked by a weight which expresses how many times the workshop participants provided input for that specific VP item. In addition, the products and services, pain creators, and gain relievers that fit the customer profile items better have been detected. The items with higher importance contribute more to the construction of the top two levels ("Summary" and "Title/Slogan") of the pyramid in Figure 2. Figure 5 shows the single case of the specific customer segment of DIHs, also demonstrating that most customer needs were covered by the HUBCAP DIH network offer. For example, the fit at the top of the figure demonstrates that the DIH ecosystem analysed is able to offer network, brokering and matchmaking to both enable other DIHs to easily find training, skills and competencies in the CPS technology/MBD domain and to improve the visibility to attract new customers. In addition, the MBD assets (tools, training, success stories, etc.) provided by the DIH network address the networking and matchmaking need to find synergies with both academic and industrial partners. Instead, the fit at the centre shows that a gain creator as an improved catalogue of services, MBD tools and knowledge triggers multiple gains in terms of an increase of knowledge and opportunity for training, an increase in the reach of MBD and specialized services, and an improvement of sales and monetization of DIH customers (SMEs). Finally, the fit at the bottom shows that a pain reliever in the form of MBD experts able to provide missing knowledge could address multiple pains for new DIHs, such as lack of skills, knowledge and technical complexity; MBD costs; and lack of confidence in MBD tools.

For the sake of completeness, the results of the VPC application for the other three customer profiles (academic partners and RTOs, technology/tool providers, and technology/tool users) are shown respectively in Figures 6–8; it can be noted that, per each type of stakeholder of the DIH network, different results have been obtained. Indeed, for academic partners and RTOs (Figure 6), the main customer jobs to be addressed by the DIH network are knowledge creation and transferring for teaching MBD, networking with MBD users, exploitation of existing MBD tools and models in applied research, and so on. To enable their satisfaction, the DIH network provides training and demonstration, community and networking services, different and customizable models and tools, and events and dissemination channels. For this kind of customer, the main pains are the effort to learn and adopt MBD, and potential adopters' lack of interest and understanding of MBD. The DIH network is capable of defusing them through effective collaboration through the sandbox, a low bureaucracy and high safety, indirect advertising, and existing teaching and training material. At the same time, this offers increased visibility, enhances the network, develops more skilled personnel and improves research quality.

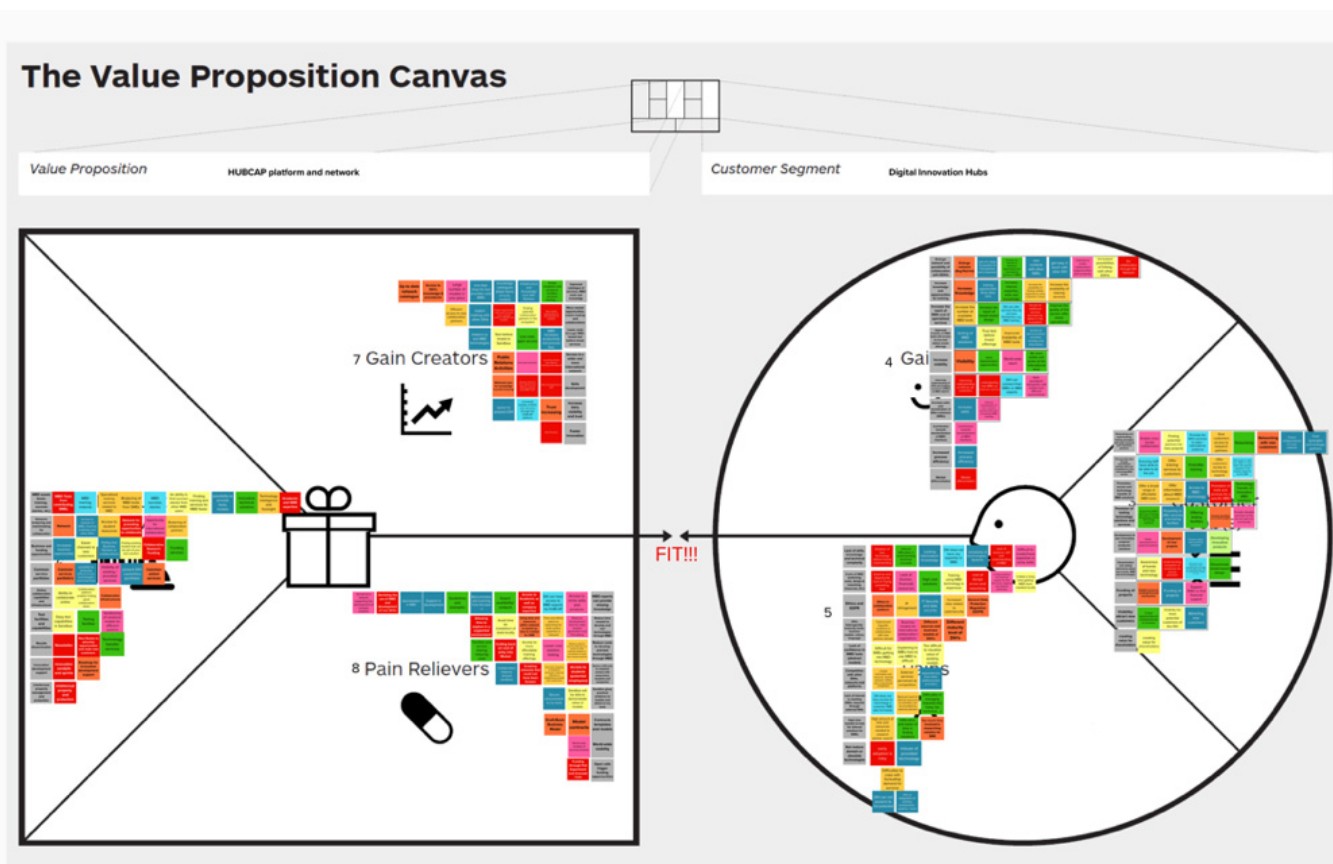

**Figure 3.** Workshop on the Mural online collaboration platform: results for VPC related to the DIH customer segments.

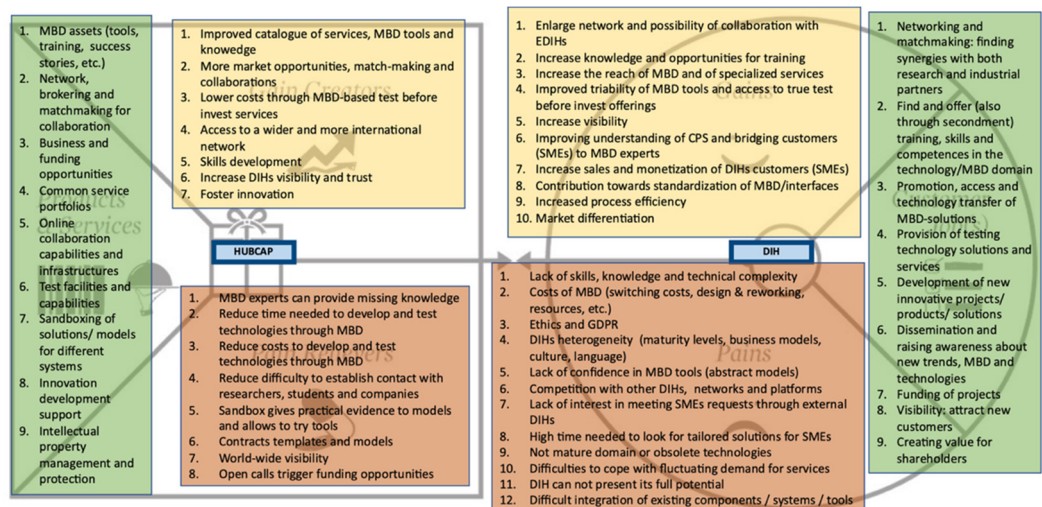

**Figure 4.** Main categories of the HUBCAP VP for the DIH customer segments.

For technology/tool providers (Figure 7), it is necessary to install, sell and provide tools and toolchains in a homogeneous environment, to attract new partners and establish network cooperation, or to provide access to a new platform with new tools, infrastructures, data and capabilities. In this case, the DIH network provides a collaboration platform as a tool repository/catalogue, a network collaboration and ecosystem service, and the dissemination of a set of best practices and success stories. Pains such as poor performance of the environment and embedded tools, a bad user experience, and a poor platform

performance can be defused by low infrastructure cost and no dependency on expensive proprietary platforms, the possibility to support customers with remote assistance and manuals, and resource use monitoring, user statistics and bug detection. Multiple gains are brought to the customer as, for example, a set of remote tools that are easily accessible, lower barriers for new users, and more visibility of new application cases and markets.

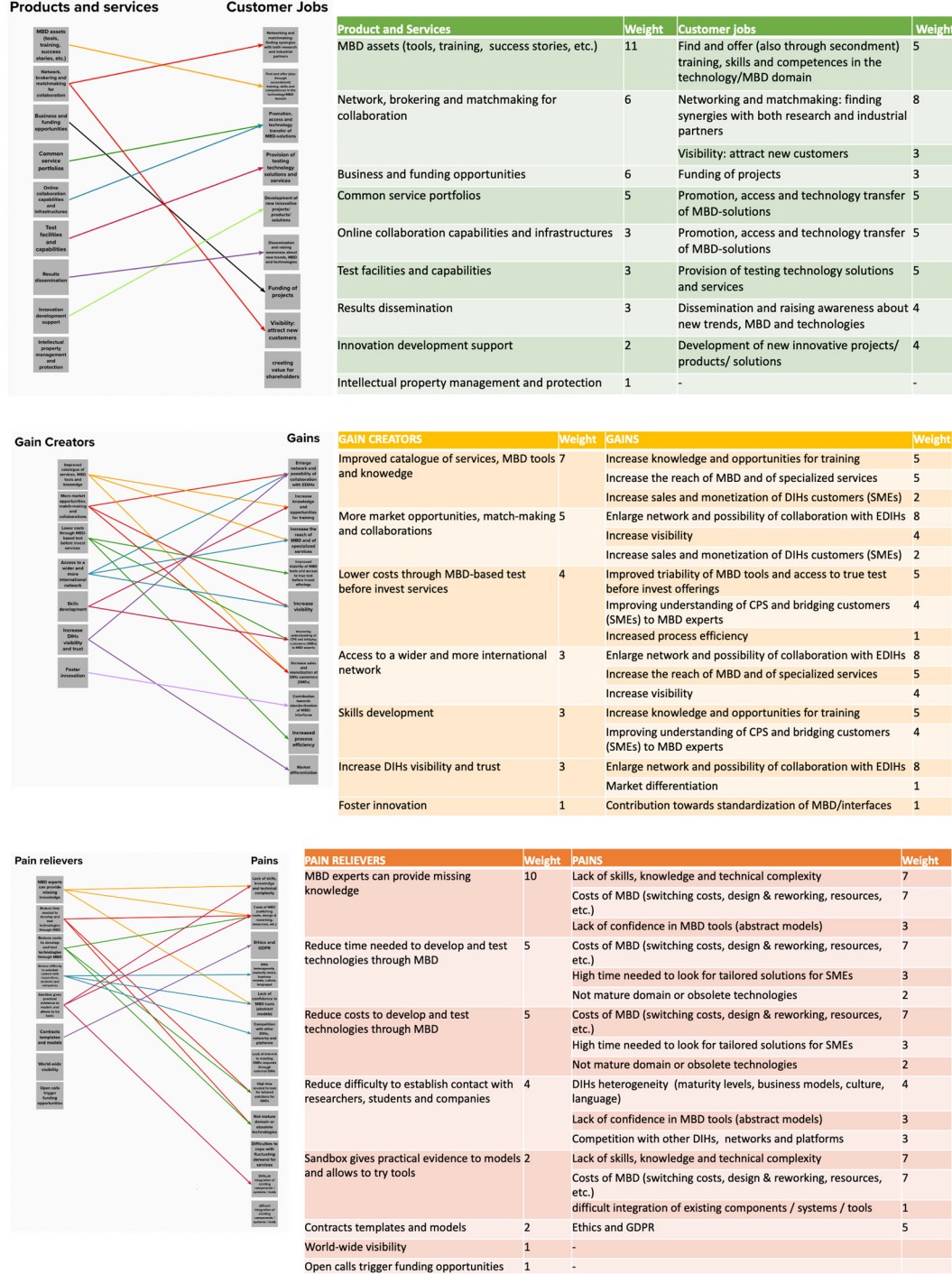

**Figure 5.** FIT for DIH customer segments among the customer profile dimensions (jobs, pains and gains) and of the VP map (products and services, pain relievers and gain creators).

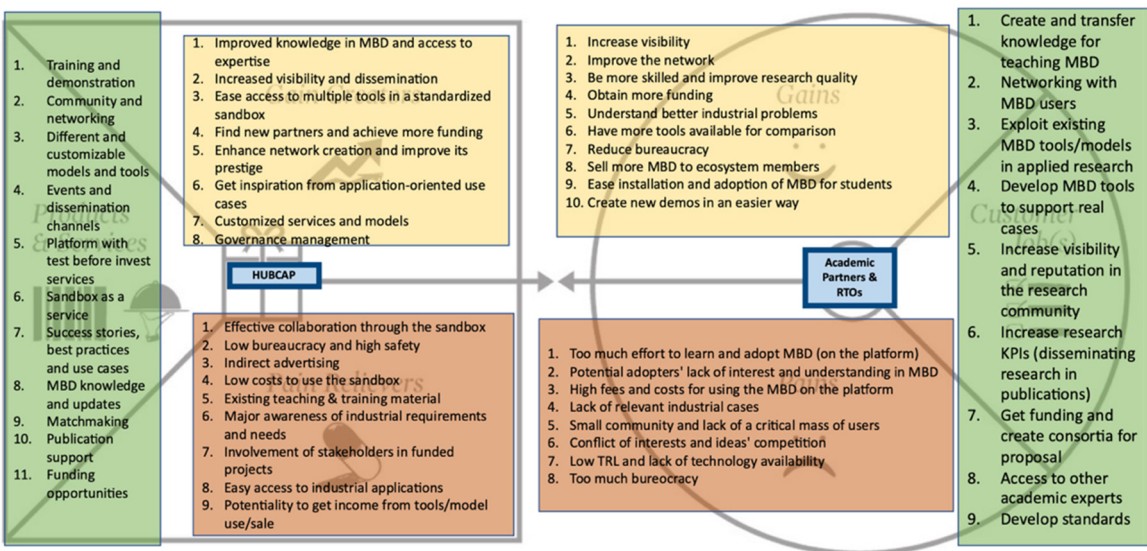

**Figure 6.** Main categories of the HUBCAP VP for the academic partner and RTO customer segments.

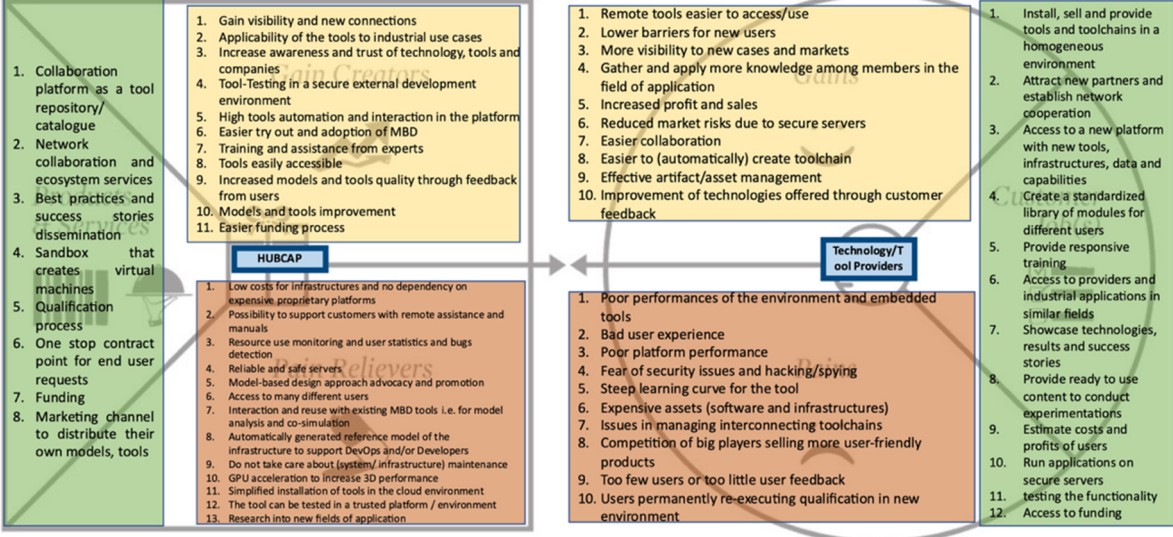

**Figure 7.** Main categories of the HUBCAP VP for the technology/tool provider customer segment.

In the case of the technology/tool users customer segment (Figure 8), the main jobs are to solve a specific problem regarding accessing new tools, prototype testing and reaching new domains, partners and networks to be more innovative and competitive. The DIH network achieves these activities by providing training knowledge and expert support, a multi-user assets catalogue to test facilities in several operating systems, and more. The main pains of this segment are IP protection, overly complex technologies/models/tools/platform/sandbox, and lack of competencies/skills/knowledge/support. The DIH network can defuse them through specialized and real-time knowledge and support, application examples for many different domains, and easy access to test technologies.

The HUBCAP DIH network, thanks to this analysis, can understand how to satisfy each of its customers, defusing pains and creating gains for each of them.

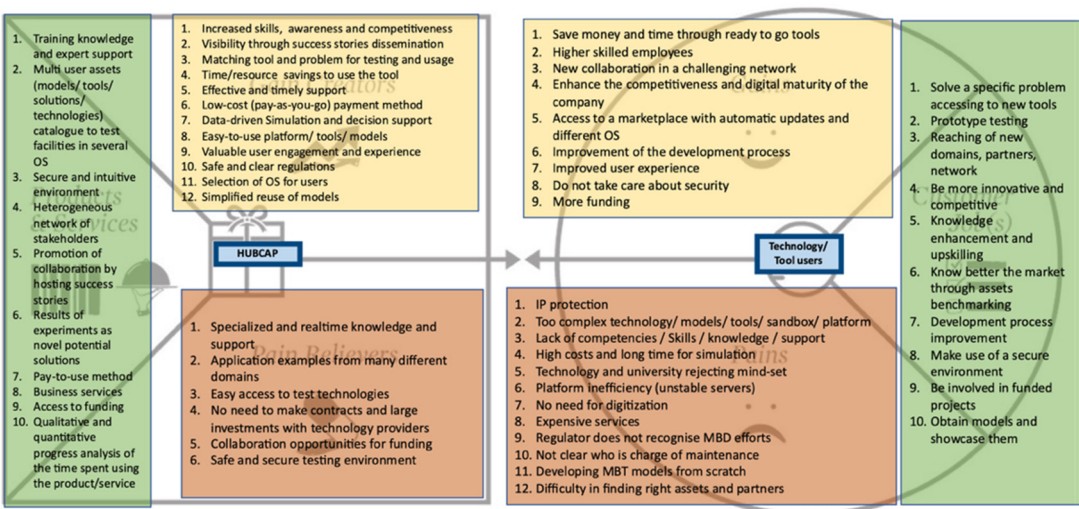

**Figure 8.** Main categories of the HUBCAP VP for the technology/tool user customer segment.

## 4. Discussion

Figure 9 shows the VPs related to the four customer segments considered in this research: the DIHs (top left), academic partners and RTOs (top right), technology/tool providers (bottom left), and technology/tool users (bottom right), useful to the HUBCAP DIH network for attracting stakeholders belonging to each of these categories; it can be noted that depending on the customers considered, the VP of the HUBCAP network changes. In particular, for the DIH customer profile, the slogan is "Leverage MBD assets to trigger a trustworthy platform-driven collaboration on CPS innovation among DIHs". In particular, in the summary, there are two main elements that HUBCAP could offer to attract this customer profile: the provision of MBD assets (tools, training, success stories, etc.) in a common service portfolio to easily test facilities and capabilities in the CPS innovation development through online collaboration infrastructures and DIHs' actions (networking, brokering and matchmaking) to provide business and funding opportunities and to manage and protect intellectual property. For academic partners and RTOs, the main slogan is "Create a recognized network of experts to spread MBD assets adoption and dissemination". Indeed, the HUBCAP offer for this customer profile in the summary is characterized by two main needs: the provision of MBD-based assets (tools and methods, training material, success stories, best practices and use cases) to foster publications through the exploitation of a sandbox as a service on a platform and the provision of matchmaking, events and dissemination channels, and funding opportunities services to support community development and networking. Concerning the technology/tools providers, the HUBCAP slogan is "Collaboration platform to address end-user requests through a sandbox providing a tool repository/catalogue". The HUBCAP network in this case offers two main elements to engage new tool providers to participate in its digital platform: a collaboration platform as a one-stop contact point to address end-user requests through a sandbox providing a tool repository/catalogue and a marketing channel to disseminate models and tools, best practices and success stories, training and funding opportunities. Finally, dealing with technology/tool users, the slogan is "Multi-user assets catalogue and related training and knowledge to test facilities in several OSs in a secure and intuitive environment through a pay-to-use method"; indeed, the strengths of the HUBCAP offer to attract new users belonging to this customer profile are, on the one hand, success stories, training knowledge and expert support provided by a collaborative and heterogeneous network of stakeholders, and on the other hand, multi-user assets (models/tools/solutions/technologies) catalogue to test facilities in several OSs in a secure and intuitive environment through a pay-to-use method.

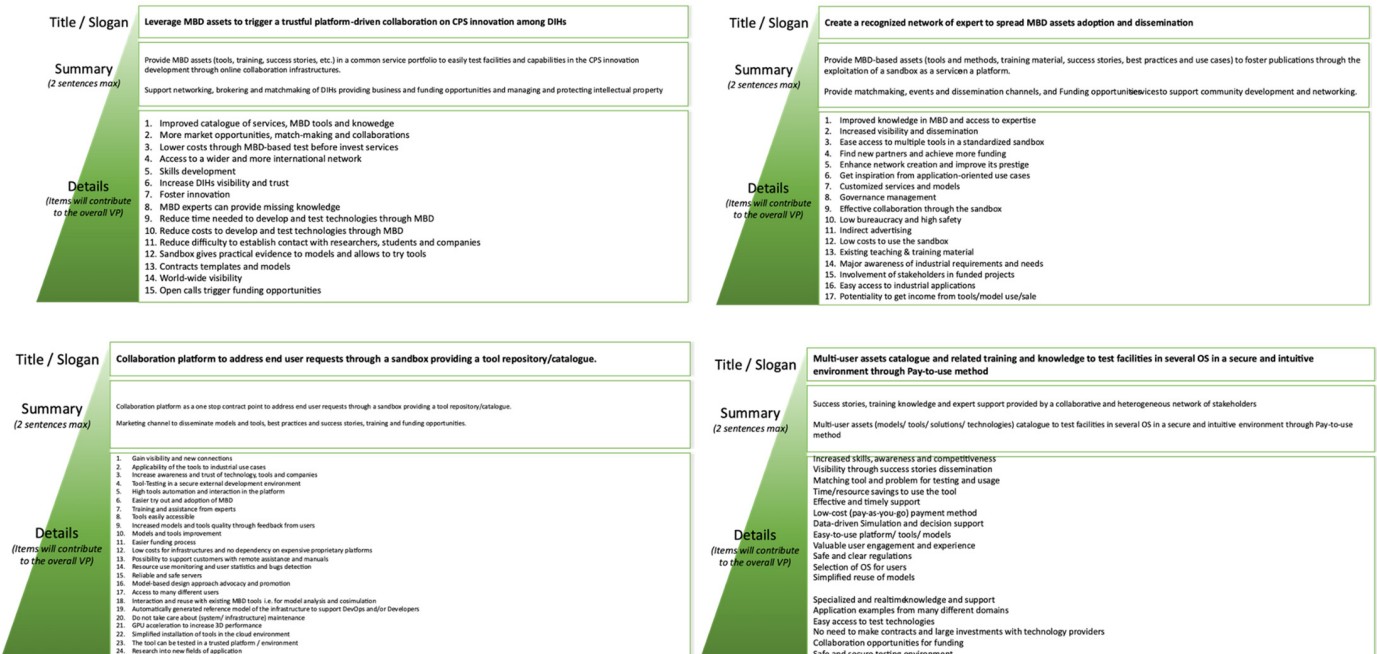

**Figure 9.** VP of the HUBCAP network for DIHs (**top left**), academic partners and RTOs (**top right**), technology/tool providers (**bottom left**), and technology/tool users (**bottom right**).

This research contributes to knowledge in multiple ways; it explores the VP items characterizing each of the four main customer profiles of a network of DIHs operating in the CPS domain and specialized in MBD and tools; it proposes a structured method to build the VP of a network of DIHs combining methods and approaches previously implemented in the business domain; it lays the groundwork to develop a sustainability plan of a network of DIHs and to develop the related business model. In the specific case of MBD models and tools provided by the HUBCAP network, this research contributes in raising the importance of MBD assets to innovate in digital technologies and impels companies to collaborate in multiple domains, putting together different and complementary competences.

From a practical perspective, this research contributes by providing a well-defined and structured VP, tailored to each customer profile, to a network of DIHs operating in the CPS domain and offering a set of MBD assets, making the network more appealing and enhancing its attractiveness to customers as well as to other potential collaborating DIH networks (because it highlights the strengths and weaknesses of its offer), as for example Circular Economy and bioeconomy [37,38]. Indeed, this research contributes to pushing MBD assets adoption to test digital technologies before actually investing in them (preventing companies from useless or bad investments) and triggers cross-sectorial and domain collaboration among multiple users through the use of the sandbox on the HUBCAP digital platform.

Finally, from a managerial perspective, the governance and managers of the DIH network offering the VP defined in this research are guided by these results to push their network in the market, develop a solid business model, make agreements with other networks, and finally achieve solid sustainability.

## 5. Conclusions

This paper, applying the VPC template, developed the VP (detailed in a three-level pyramid) of the HUBCAP DIH network aiming to support European SMEs in the adoption of MBD methods and tools to develop CPSs. In the analysis, four different customer segments were considered (i.e., DIHs, academic partners and RTOs, technology/tool providers, and technology/tool users). The research proposes the VPC approach to obtain and prioritize the VP items per customer segment; then, it develops a VP pyramid, structured

on three levels (title, summary, details), which is representative of the DIH network offer and appeals to attract stakeholders belonging to its different customer segments. Going through the three levels with a bottom-up approach (design vision) helps the organization by offering the VP to build it in a more structured way. Instead, going through a top-down approach (customer vision) allows the organization to catch more effectively the customers to make them adhere to the VP. Indeed, the mission of the VP statements gathered in this template is to create visibility for the HUBCAP network (constituted by DIHs and services and MBD assets providers in the CPS technology domain across Europe) to allow SMEs access to this set of assets. Therefore, this research sought not only to test the VPC in the DIH ecosystem domain but also to propose a more complete approach to systematize its results in a VP pyramid. The results show that depending on the customers considered, the VP of the DIH network changes. Notwithstanding the novelty of this research in defining a structured VP of a network of DIHs, limitations can also be described. Indeed, in this paper, only the results related to the HUBCAP network (which represents a niche in the CPS domain) are presented. The same analysis could be performed for different networks operating either in the same domain, the CPS, or others (e.g., artificial intelligence). A comparison among the value propositions of these different networks could be performed to unveil the peculiar characteristics of each of them and detect the touch points among them to encourage future collaborations. Once the collaborations among different networks are materialized, the overall VP can be defined, providing room to bolster the actions of future European DIHs. Finally, this analysis represents only the first step towards the sustainability project of the HUBCAP network of DIHs. For each customer considered, a well-defined business model will be developed. Together, an analysis of the network assets (MBD and tools, competences and services) will be performed. The service offering will be analysed, detecting the services that contribute more to creating revenues among those constituting the HUBCAP service portfolio (decoded through the Data-based Business-Ecosystem-Skills-Technology [D-BEST] reference model); it must be highlighted that in the HUBCAP VP, a key role is played by the MBD assets (models and tools) because they are an effective means for testing before investing and thus are able to trigger along the CJ a high quantity of connected services to address the digital transformation.

**Author Contributions:** Conceptualization, C.S. and S.T.; methodology, C.S.; validation, C.S.; formal analysis, C.S.; investigation, C.S.; data curation, C.S.; writing—original draft preparation, C.S.; writing—review and editing, C.S.; visualization, C.S.; supervision, S.T. All authors have read and agreed to the published version of the manuscript.

**Funding:** This research was funded by the European Union's Horizon 2020 research and innovation programme under grant agreement No 872698 (HUBCAP Innovation Action).

**Informed Consent Statement:** Informed consent was obtained from all subjects involved in the study.

**Conflicts of Interest:** The authors declare no conflict of interest. The funders had no role in the design of the study; in the collection, analyses, or interpretation of data; in the writing of the manuscript; or in the decision to publish the results.

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
