# Peer review of "Building the Value Proposition of a Digital Innovation Hub Network to Support Ecosystem Sustainability"

_sustainability, doi:10.3390/su141811159_

Round 1

Reviewer 1 Report

Interesting, well written paper. No special comments.

Author Response

Thank you for appreciating our research.

Reviewer 2 Report

Dear Authors

After reading your article I would like to propose the following changes:

1. Create a literature review section separate from the introduction. There are contents in section 2. that should be transferred to the literature review;

2. the conclusions belong to you authors and therefore should not cite other authors. For that you have the discussion section;

3. There are papers from 2021 and 2022 that should be cited in your paper.

Good luck with the next step.

Best Regards 

Author Response

Dear Authors

After reading your article I would like to propose the following changes:

  1. Create a literature review section separate from the introduction. There are contents in section 2. that should be transferred to the literature review;

A literature review sub-section has been added in section 2 (“Research context”) as sub-section 2.1. Its aim is to introduce DIHs and their role in the SMEs digitization.

  1. the conclusions belong to you authors and therefore should not cite other authors. For that you have the discussion section;

Thanks for your suggestion. References have been deleted from conclusions section.

  1. There are papers from 2021 and 2022 that should be cited in your paper.

New references have been added.

Good luck with the next step.

Best Regards 

Thank you for taking the time to go through our manuscript and providing your precious comments to enhance its quality.

Round 2

Reviewer 2 Report

Dear authors,

In my last review I proposed a section Literature Review. However, you creat a section 2. Materials and Methods. But. This section it’s Literature Review.

Please, see carefully all cited references.

best regards 

Author Response

Sorry for the oversights.

Section 2 ("Materials and Methods") has been renamed as "Literature Review".

References have been revised.